# Stream Slope as an Indicator for Drowning Potential at Low Head Dams

Jason W. Poff [1],* and Rollin H. Hotchkiss [2]

1    Department of Civil and Construction Engineering, Oregon State University, Corvallis, OR 97331, USA
2    Department of Civil and Construction Engineering, Brigham Young University, Provo, UT 84602, USA
*    Correspondence: poffja@oregonstate.edu; Tel.: +1-150-3724-8687

**Abstract:** With the increasing availability of low head dam inventories for the United States, the next challenge is discovering how to determine what dams pose the greatest risk to public safety, preferably before a death occurs. Submerged hydraulic jumps create the dangerous current that drowns roughly 50 recreationists each year, and high tailwater is a key element in its formation. Using a simplified approach based on the Manning equation, flat downstream slopes can be a predictor of high tailwater. Stream slopes at low head dams in Colorado, Idaho, Indiana, Maryland, New Mexico, North Carolina, and Pennsylvania were collected from the NHDPlus HR, and dams with recorded fatalities were compared to stream slopes at low head dams with no recorded fatalities. Using the Mann–Whitney U test, there was not enough evidence to reject the null hypothesis that there is no statistically significant difference between the two populations. Until more fatality data are compiled and more low head dam locations are verified, individual testing of dams is recommended to establish each respective flow range that is likely to pose a risk to public safety.

**Keywords:** low head dam; stream slope; public safety; hazard inventory; submerged hydraulic jump; NHDPlus HR; Mann–Whitney U test

## 1. Introduction

### 1.1. Purpose of the Work

#### 1.1.1. Background

Low head dams, often deceptively innocuous in appearance, have claimed over 1000 lives in the United States [1]. Given the nickname drowning machine, the American Society of Civil Engineers estimates 50 people drown each year, caught by currents at the face of these dams [2]. Low head dams are also called weirs, run-of-the-river dams, diversion dams, or low overflow structures. These structures are built mainly to divert water for irrigation or mills (see Figure 1) [3]. Commonly found all across the United States, experts estimate there are tens of thousands [1].

These dams can be dangerous because they have the potential to create a condition called a submerged hydraulic jump. The submerged hydraulic jump creates a reverse roller current, where the water near the bottom of the stream is moving downstream, but the water at the top is being pushed back upstream towards the face of the dam. A person caught in this current will be repeatedly forced under, float up, and be brought back into the face of the dam in a vicious cycle. Low head dams are not regulated by the federal government or most state governments because they fall below jurisdictional size categories [4]. These dams are not very dangerous if they fail due to very small water storage volumes, but it is safety at and around low head dams that matter.

Not all low head dams are capable or likely to produce a submerged hydraulic jump, and even the ones who can, will not be dangerous year-round: it all depends on flow and stream conditions. People might be lured into a false sense of security around these dams as a result [5]. There is a need to be able to predict the danger of low head dams generally

so regulators and owners can focus on protecting the public from the most dangerous dams first, and by doing so save more lives. Researchers and professionals have gone to great efforts to gather low head dam inventories across the United States, and the next step is to encourage and take action using that data [1]. This paper attempts to develop a quick and easy way for stakeholders to characterize the potential danger of low head dams using stream slope.

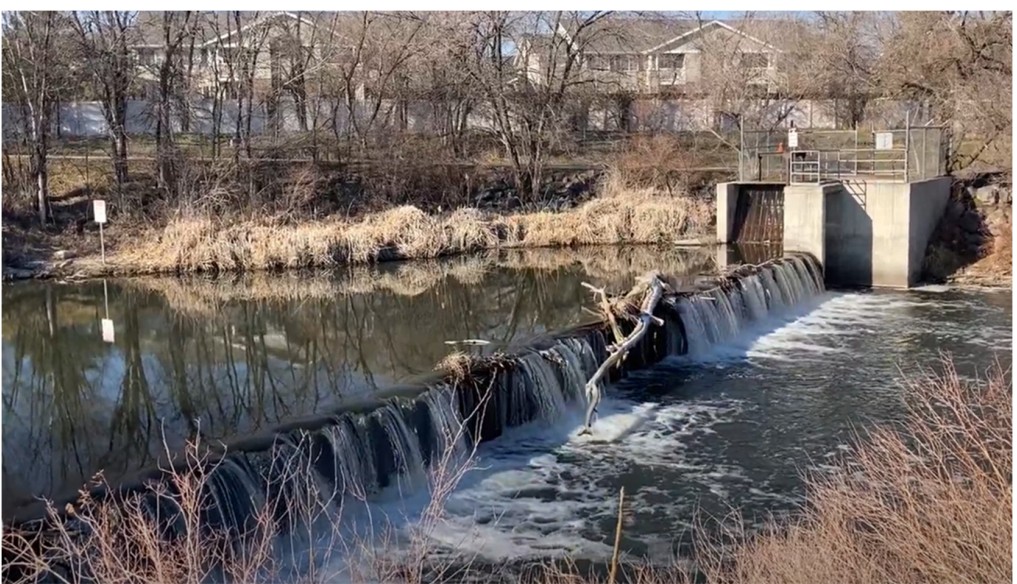

**Figure 1.** Low head dam used for diversion in Taylorsville, Utah. A site with a single fatality.

1.1.2. Proposed Theory

The theory is that submerged hydraulic jumps are more likely to occur at low head dams when the stream slope downstream of the dam is flat. A hydraulic jump is submerged when the tailwater (water level downstream of the dam) is higher than the sequent depth associated with the jump. Using a simplified approach based on the Manning equation, a flat slope can be a predictor of high tailwater. The Manning equation is:

$$Q = \left(\frac{1.00}{n}\right) A R^{\left(\frac{2}{3}\right)} \sqrt{S} \tag{1}$$

where $Q$ is flow rate (m$^3$/s), $A$ is flow area (m$^2$), $R$ is the hydraulic radius (m), $n$ is the Manning's roughness coefficient, and $S$ is the channel slope (m/m). Solving for flow per unit length (denoted here as $Q/b$, in m$^3$/s·m) and assuming a wide rectangular channel we obtain:

$$\frac{Q}{b} = (1.00)\frac{y^{\left(\frac{5}{3}\right)}\sqrt{S}}{n} \tag{2}$$

where $y$ is tailwater depth (m). Given a steady increase in flow per unit length at any given dam, those with flat slopes (very low $S$ values) will have a much higher tailwater ($y$) than the same flow at other slopes. High tailwater increases the likelihood of a submerged hydraulic jump. This method does not account for other factors that may produce high tailwater, such as scour trenches and debris deposition downstream from the dam.

If there is a connection between stream slope and drownings at low head dams, public safety officials will be able to easily identify and provide evidence that certain low head dams are more dangerous and should be either modified, protected with gates and signs, or removed.

*1.2. Literature Review*

1.2.1. The Danger of Low Head Dams

A hydraulic jump is a phenomenon that occurs when water that is moving at a high velocity drops into a zone of lower velocity, and there is an abrupt rise in the water surface. Low head dams create this shallow, fast-moving water (supercritical flow) as the water plunges off the face of the dam, and merges with the deep, slow-moving water below (subcritical flow). A jump is also evidenced by reverse circulation of the flow on the surface, otherwise known as countercurrent velocities [5].

The hydraulic jump can become submerged at low head dams under high tailwater conditions. The backflow of water on the surface of the hydraulic jump travels upstream towards the face of the dam. Combining with the fast downstream current on the bottom, an individual who goes over the dam or moves close enough to it on the downstream side will be caught in a vicious circular flow (see Figure 2). Buoyancy is also reduced by the entrained air from the dam [6]. Even with a personal flotation device, these powerful currents can drown the most experienced swimmer [4].

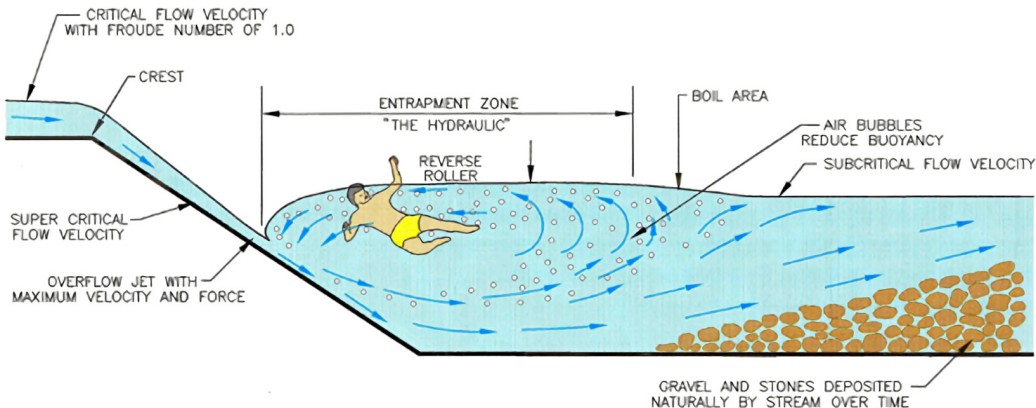

**Figure 2.** Elevation view of a submerged hydraulic jump at a low head dam (Reprinted with permission from Wright et al. [7] published by the Association of State Dam Safety Officials, 1995).

In certain circumstances, submerged hydraulic jumps can also form due to streambed erosion. When fast-moving water carries rocks and sediment and deposits them downstream, a scour hole may be formed, backed by a large deposit of sediment (see Figure 2). This can create a high tailwater condition, which causes the hydraulic jump to become submerged. This phenomenon can occur in areas without low head dams, as was the case on the Jordan River in Murray, Utah. In this instance, a slab of concrete was placed in the streambed to protect pipes from erosion, but the water ended up eroding the streambed directly downstream, leading to the formation of a scour hole and a submerged hydraulic jump. Two kayakers who were unaware of the hazard drowned as a result [8]. This circumstance is not treated in this paper.

1.2.2. Other Methods to Characterize Hazards at Low Head Dams

There are many ways to characterize the danger potential at specific low head dams. A method developed by Leutheusser and Fan from experimental data quantifies the velocity of the reverse current and concludes that the velocity is highest when the jump is just mildly submerged. The velocity decreases as the tailwater depth increases, until the submerged nappe flips to the free surface and the jump is drowned out [9]. Tony Wahl uses the same method presentenced by Leutheusser and Fan with one modification to create a spreadsheet that calculates whether a submerged hydraulic jump is likely to occur and the countercurrent velocity for a particular dam given the following inputs: flow range, weir length, structural height above downstream invert, bed slope, Manning's n parameter, and channel width [5]. The spreadsheet was tested with a case study used by Devadason

and Schweiger in a computational fluid dynamics (CFD) model of the Dock Street Dam in Pennsylvania [10]. The range of dangerous flows calculated by the model and spreadsheet were very similar [5].

With experimental data, Olsen created ranges for a simple parameter called the "risk factor" (the depth of the headwater upstream of the dam minus the depth of the tailwater downstream of the dam divided by the height of the dam) that could predict if the dam had a swept-out jump, submerged hydraulic jump, or a drowned out jump. If the risk factor was in the range that indicated a submerged hydraulic jump, it was considered high-risk, but if it was above or below that (indicating a swept-out jump or drowned out jump) it was considered low-risk [11].

### 1.2.3. National Hydrography Dataset (NHD) plus High Resolution and Stream Slope

The United States Geological Survey (USGS) created a tool called the NHD plus high resolution (HR), which is a geospatial hydrography framework packed with information. It combines three datasets: the National Hydrography Dataset (which has a scale of 1:24,000 or better), the nation-wide Watershed Boundary Dataset, and the 10-m ground spacing 3D elevation program digital elevation model (DEM). This is a new version of a previous framework, the NHDPlus version 2, and it brings the scale down to the neighborhood level. The NHDPlus HR has a highly curated, interconnected stream network with average flow values, stream direction, and many other features [12]. The NHDPlus HR allows for large-scale experiments, such as the modeling of the Mississippi River Basin at a continental scale, as part of the National Flood Interoperability experiment. In this study, Tavakoly et al. used the high-resolution river data from the NHDPlus HR to test the routing application for parallel computation of discharge (RAPID) model and evaluated its performance in relation to topography and major dams, using the variable infiltration capacity (VIC) model as an input. They also compared the simulated stream flow results obtained using the RAPID model with and without the VIC input [13].

In 2017, researchers Cohen, Wan, Islam, and Syvitski worked on a large-scale stream slope map [14]. They took a global 15 arc-second (450 m) digital elevation model (DEM) from the USGS (for latitudes above 60° N, a 1 arc-min DEM from etopo was used) and stream network data from the USGS shuttle elevation derivatives at multiple scales (HydroSHEDS). Adding a 50 km cap to the segment length, they assigned elevations from the underlaying DEM as the highest and lowest elevation from anywhere on the stream segment. Researchers assigned the slope by dividing the change in elevation over the segment length. Comparing with 34 observed slopes around the world, they found that the up-scaled 6 arc-min map had the best correlation. The $R^2$ value was 0.64, and it had a RMSE of 0.0016. Lower slope values were consistently overpredicted, so an adjustment equation was developed by linear regression to reduce this bias. The method developed by the researchers was found to be more accurate for calculating stream slope on a large scale (drainage areas greater than 1000 km$^2$) than the previous version of the National Hydrography dataset, NHD Plus version 2.0 [14].

There is a newer dataset called the National Hydrologic Geospatial Fabric (hydrofabric) for the next generation (NextGen) hydrologic modeling framework. Similar to the NHDPlus HR, it represents the drainage network for the whole United States and calculates the slope for each flow path. It is used to execute the NOAA next generation (NextGen) water resource modeling framework [15]. At the time of the research, this dataset was not available, and thus has not been tested.

### 1.3. Gap in the Research

This paper aims to move towards a risk-based analysis of the dangers at low head dams. The next step after an inventory of the locations of low head dams is a characterization of their danger potential. The proposed method is simple, automated, and easily scalable to the whole United States. This would be extremely useful as a tool to not only provide

information to decision makers, but to raise awareness of the danger of low head dams in each community.

## 2. Materials and Methods

### 2.1. Materials

#### 2.1.1. Fatalities at Low Head Dams

A fatality database can be used as a list of dams that almost certainly create submerged hydraulic jumps. Brigham Young University maintains a national low head dam fatality database with 625 fatalities recorded at 315 different dams [16]. It is certainly incomplete, but it is the largest public database available. The data was collected by former graduate students at BYU, Dr. Bruce Tschantz, Charlie Walbridge from the American Whitewater Association, and others.

#### 2.1.2. Low Head Dam Inventory

The National Low-head Dam Inventory task force [1] has collected and quality checked low head dam locations for eight states as part of this study: Colorado, Idaho, Indiana, Maryland, New Mexico, North Carolina, Pennsylvania, and Wyoming (see Figure 3). There are more data for other states, but they are either incomplete or not thoroughly checked by engineering professionals with experience working with dams. Following the removal of low head dams from each of the eight states that had fatalities recorded at them in the BYU database, this data can be used as the low head dam sites that are not known to create submerged hydraulic jumps, although they very well could. In comparing the two datasets, the hope is that there is a significant enough relationship between the stream slope and recorded fatalities that it will be evident even with some low head dams that create submerged hydraulic jumps in the non-fatal dataset.

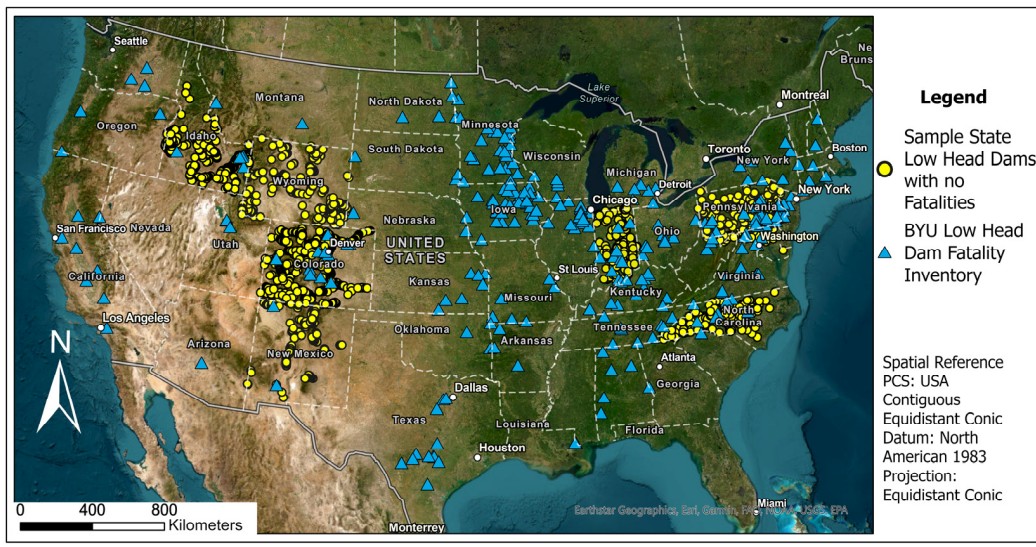

**Figure 3.** Map of all of the low head dam locations used in the study.

The low head dam inventory compiled by the National Low-head Dam Inventory task force is collected in different ways by each state. Some states, such as Idaho, New Mexico, and Wyoming were made entirely by student researchers at BYU combing through every river in the state on Google Earth. These datasets are then sent to practicing engineers who quality check all of the points and make the final decisions on what is likely a low head dam or not. Colorado, Indiana, Maryland, North Carolina, and Pennsylvania low head dam inventories were provided by state government agencies, such as the Department of Natural Resources, Department of Environmental Quality, or the Dam Safety Divisions of the Department of the Environment.

2.1.3. Stream Slope

Slope data were developed by the United States Geological Survey (USGS) as a step in making the National Hydrograph dataset (NHD) [12].

The slope along each flow line is calculated using the 10 m DEM of the United States. Hydrologists place a point at each beginning, intersection, or end of a flowline (connecting to a lake or ocean). From the top of the stream network (streams with order 1 and so on) down, they calculate the catchment area for each point. Within each area, the lowest elevation from the 10 m DEM is assigned to the point. This method allows for some error between the DEM and stream network because even if the elevation closest to the catchment point is incorrect or not located on the river itself, the system will still use the lowest point in the area, which is better than using the closest elevation value. Once the elevation has been assigned to each point, the slope of the stream segment can be calculated by taking the difference in elevation between the beginning and end of the segment and dividing it by the length. The slope is then assigned to the stream segment as a unitless value [12]. Short stream segments are more sensitive to inaccuracies in the DEM [14], because there are not as many DEM values in these smaller catchment areas. If in these few there is one that is inaccurately low, then the inaccuracy would be amplified by dividing it only by a short length. This error could happen for longer segments, but by dividing it by a longer length the same order of magnitude error would be less potent than for a short length. For example, an error of 1m in elevation would have a larger effect over a 20 m length than a 100 m length. For the purposes of this study, the slope downstream of the low head dam is needed. This method could also be slightly inaccurate if one of the catchment points is located near enough to a dam and completely upstream of it so that the water level elevation is impacted by the dam significantly. This would give a higher elevation drop than is true to the general slope of the stream if the other catchment point is downstream of the dam, with the error increasing the closer the point is to the dam (see Figure 4). In this first analysis of its kind, no accuracy tests were performed on the derived slope data.

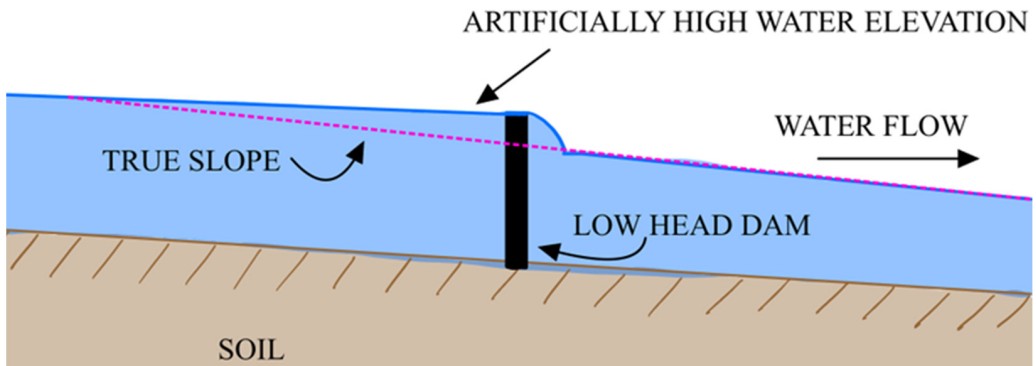

**Figure 4.** Elevation view of a potential error in water surface slope calculation.

The accuracy of the stream slope calculated using the NHDPlusHR method should be carefully considered. In the study by Cohen et al., root mean square error tests were conducted on the previous version of NHDPlus (which used 30 m DEM data instead of 10 m) using 25 real observed stream slopes from across the contiguous United States [14]. While this is not the same dataset used in the current research, it provides a starting point for evaluating the accuracy of the NHD method. The NHDPlus version used in the Cohen et al. study had a correlation value of 0.48 ($R^2$), a root mean square error of 0.0078 (RMSE), and a difference of averages of 0.0017 (estimate average–observed average, 0.003–0.0012). It is thought that the NHDPlus version 2 slope values may be impacted by the use of short stream segments due to its detailed stream network, which can be more sensitive to the DEM vertical resolution and biases [14]. Both Cohen's global slope map (GLoRS) and NHDPlus tend to overestimate the slope, particularly at smaller values. To address this

issue, an adjustment equation, such as the one used by Cohen, could potentially be applied to the NHDPlusHR after further testing [14].

*2.2. Methods*

2.2.1. National Stream Slope Map

The NHDPlus HR can be accessed and downloaded from The National Map, a web map by the USGS that delivers a variety of topographic information for the United States. At the time of the analysis, only the HU-4 subregion extent data were available to download. There are 227 subregions, and they can all be downloaded at once by downloading a txt or csv file of all 227 download links and using an open-source download manager called "uGet", as recommended by the USGS. Each subregion has a geodatabase containing all of the information for that area. Within that geodatabase, there are 3 folders: "Hydrography", "NHDPlus", and "WBD". Within the hydrography folder, there will be a shapefile called "NHDFlowline" that can be added to the map in a GIS program, such as ArcGIS Pro or QGIS. This contains every flowline for that region, but slope is not an attribute for those lines. Some attributes, including slope, are separated into another table in the WBD folder called "NHDPlusFlowlineVAA", where "VAA" stands for value-added-attributes. Each flowline has a unique code called a "Reachcode", and the attributes in the NHDPlusFlowlineVAA table can be added to each flowline by joining the table to NHDFlowline by the unique Reachcode for each. This was achieved using a Python script and the package arcpy to loop through all of the subregions and add slope to each flowline. An example of the stream slope map created is shown for Harrisburg, Pennsylvania in Figure 5.

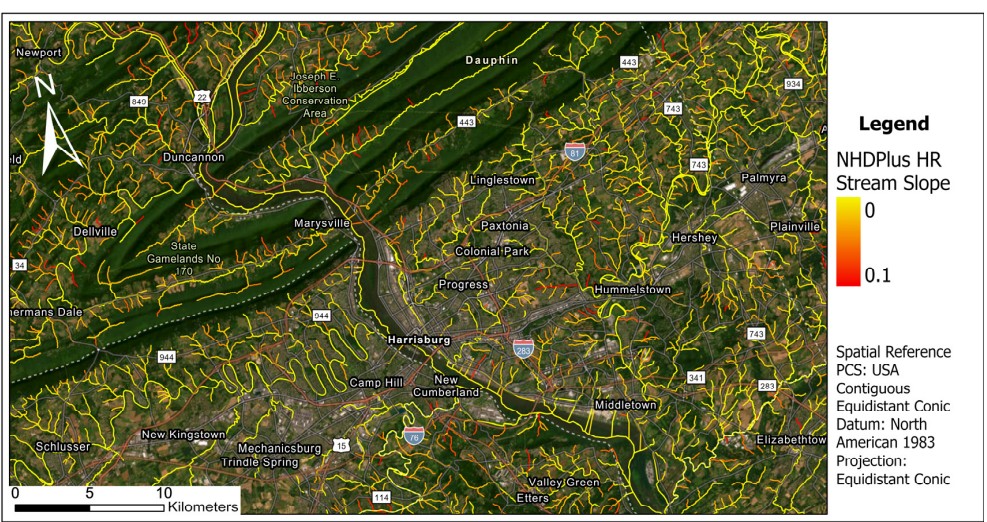

**Figure 5.** Stream slope map of the area around Harrisburg, Pennsylvania.

For ease of working with the data, each subregion was combined with others in its region. There are 21 total regions, but only 1-18 were used because they cover the contiguous United States, where this study is focused. It is possible to combine all of the regions into one large shapefile, but for ease of computing and avoiding large processing times, they were combined into three large shapefiles based on the region: 1–7, 8–14, and 15–18.

2.2.2. Attaching Slope to Low Head Dams

The next step was to combine each point representing a low head dam or a low head dam with a fatality to the nearest stream and its slope. One km buffers were created around each low head dam point, and the stream slope maps were clipped with this buffer to eliminate stream data that would not be used. The nearest stream slope value was added to the low head dams by using the spatial join tool in ArcGIS Pro. A few points from some of the low head dam state inventories had all null values in their attribute fields, and so these

were deleted. A few other low head dam points were assigned slope values of −9998 (0 to 5 points per state), which means that the USGS was unable to calculate a stream slope with their method. Each point was examined, and another stream segment was selected that had a calculated slope that was representative of the stream slope downstream of the dam. The line had to be covering the dam without a nearby catchment point on the upstream side of the dam or start downstream of the dam by about a least 10 m.

The closest line representing the streams to each low head dam point does not necessarily represent the correct stream that the dam is on. Low-head dams often have diversions, and if the line for the diversion happens to be closer to the point than the line for the stream with the dam, the slope for that diversion canal will be added to the dam instead of the actual stream slope (an example is shown in Figure 6). Every low head dam point with a fatality was checked for selection accuracy in each of the testing states (CO, ID, IN, MD, NM, NC, and PA) except Wyoming, because there are no recorded fatalities there in the BYU database. Any selection errors were fixed. When there were selection errors, the slope selected was almost always higher than the slope of the correct line. This was because either the diversion stream or another tributary was selected.

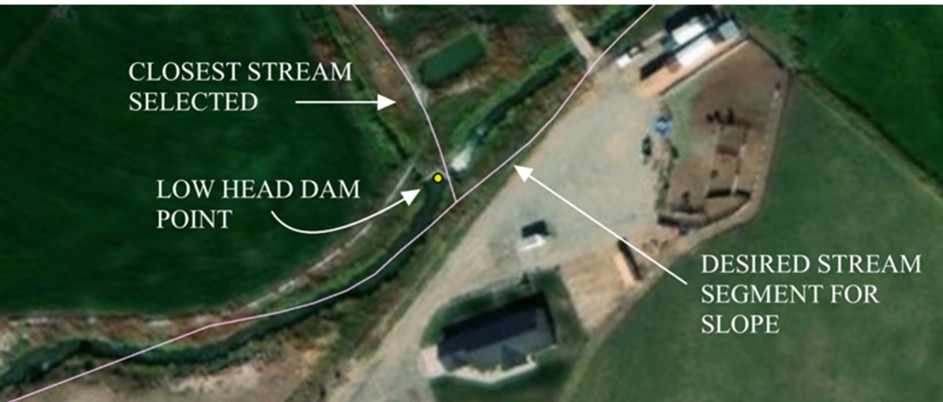

**Figure 6.** Example of a stream selection error for a low head dam with no fatalities in Idaho.

Not every low head dam point with no fatality recorded could be checked individually on account of the volume of data points, so only the outliers for each state were checked. This was justified because when going through all points with fatalities, the error was almost exclusively an uncharacteristically steep slope. The outlier limits were determined by adding 1.5 times the interquartile range (IQR) to the 3rd quartile value for the upper bound. No lower bound existed because the 1st quartile minus 1.5 times the IQR was below 0 for each state. No slopes below 0 could be calculated using the NHDPlus HR method [12]. Every low head dam that had a slope equal to or higher than the state outlier limit was checked for selection accuracy and manually changed if needed to a segment that better represented the stream slope downstream of the dam, according to the same guidelines as picking new stream segments for the no slope values. Once all outliers were checked, the outlier limit was recalculated and any new dams that were now outliers were checked, and so on. In all states, the outlier limit barely changed when recalculated.

### 2.2.3. Statistical Testing

The Mann–Whitney U test was selected to determine if there was any statistical significance to the difference in the distributions between stream slope at low head dams with recorded fatalities and at low head dams with no recorded fatalities. It was selected because the assumptions best fit the data. A Mann–Whitney U test is a nonparametric test that uses ordinal ranking for hypothesis testing with two independent samples. The test compares the sample medians, and if there is a significant difference (for this experiment, a significance level less than 0.05) there is a high likelihood that the samples represent different populations with different medians [17]. There are several assumptions that need

to be met: first, each sample has been randomly selected from the population it represents. This assumption is met conditionally. Narrowing the research question to the relationship between the stream slope and dams with fatalities to just the states where data are available (CO, ID, IN, MD, NM, NC, and PA), and assuming that the low head dam inventories (fatal and non-fatal) are complete, then this condition is met because the whole population is being tested. It will still be useful information for these states to know individually, because they can use the conclusions to start looking at modifying or removing dams in their inventories with flatter stream slopes first. If more data is added, especially dams with fatalities, these tests should be performed again. The second assumption is that the two samples are independent of each other. One stream slope at one low head dam does not affect the stream slope at another, so this condition is met. Third, the variable being observed is a continuous random variable. A continuous random variable is a type of random variable that can take on any value within a specified range, and stream slope fits perfectly. The last assumption is that both underlaying distributions are identical in shape. This condition is met generally and in all states with enough fatal low head dam sites to give the distribution any shape [17].

The Mann–Whitney U test was carried out using SciPy, a free and open-source Python library commonly used for scientific computing [18]. Tests were carried out for each individual state, and all of the sample states combined. All tests were two-sided, using the asymptotic method for calculating the p-value. Two-sided tests were performed because while it was hypothesized that the median slope of the low head dams with no recorded fatalities would be higher than the dams with fatalities, it is important to know if it is the other way around.

## 3. Results

### 3.1. Distribution of Stream Slopes at Low Head Dams

3.1.1. All Sample States (CO, ID, IN, MD, NM, NC, and PA)

Figure 7 shows the distribution of all the stream slope values at low head dams with a recorded fatality for all of the sample states: Colorado, Idaho, Indiana, Maryland, New Mexico, North Carolina, and Pennsylvania. All stream slope values greater than 0.02 m/m are included in the overflow bin at the far right. A slope of 0.02 was chosen because that was the upper outlier limit found for low head dams with no recorded fatality in the sample states (see Figure 8). All bins are 0.0005 m/m in width.

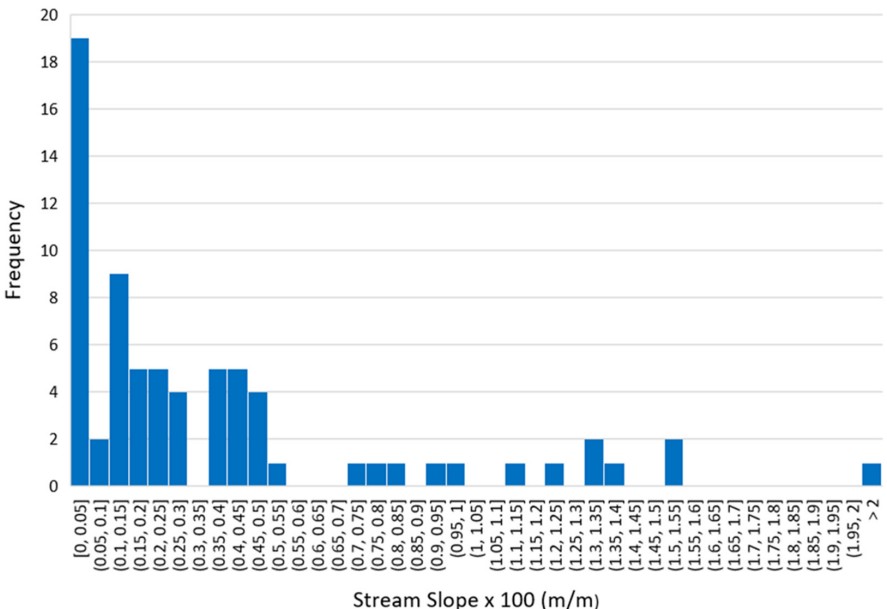

**Figure 7.** Stream slope at low head dams in sample states with fatalities recorded.

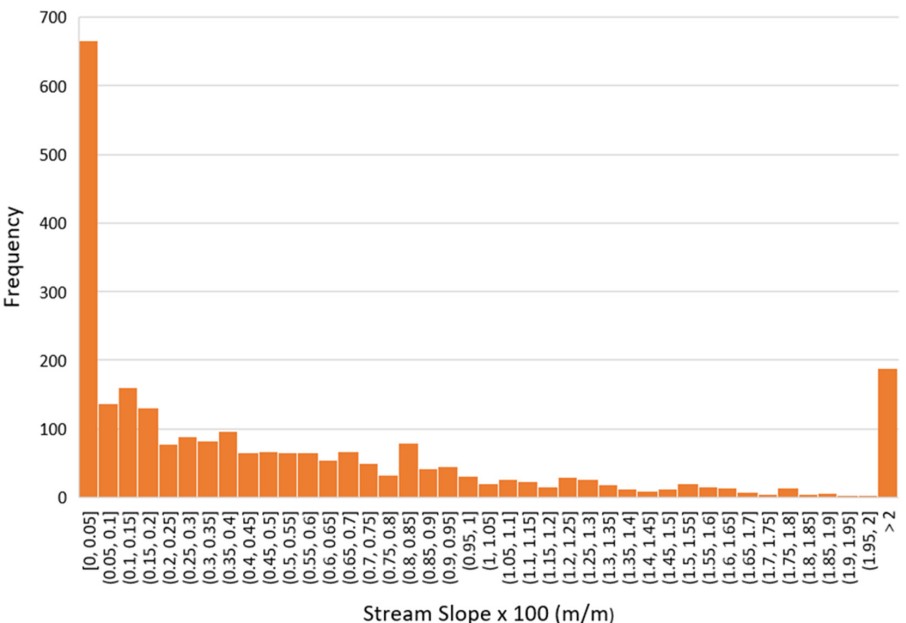

**Figure 8.** Stream slope at low head dams in sample states with no fatalities recorded.

The distribution of all low head dams with no recorded fatality in the sample states is shown in Figure 8. It has the same general shape as Figure 7, with a massive majority in the first column from 0 to 0.0005 m/m and a heavy right skew. The only major difference between the two distributions is that low head dams with no recorded fatalities has a much longer tail. This is seen clearly in Figure 9. Figure 9b shows the whole range of values in a box and whisker plot, while Figure 9a narrows in on the difference between the quartiles and mean of the distributions. The low head dams with fatalities dataset median is about 0.0021 m/m and low head dams without fatalities is about 0.0032 m/m.

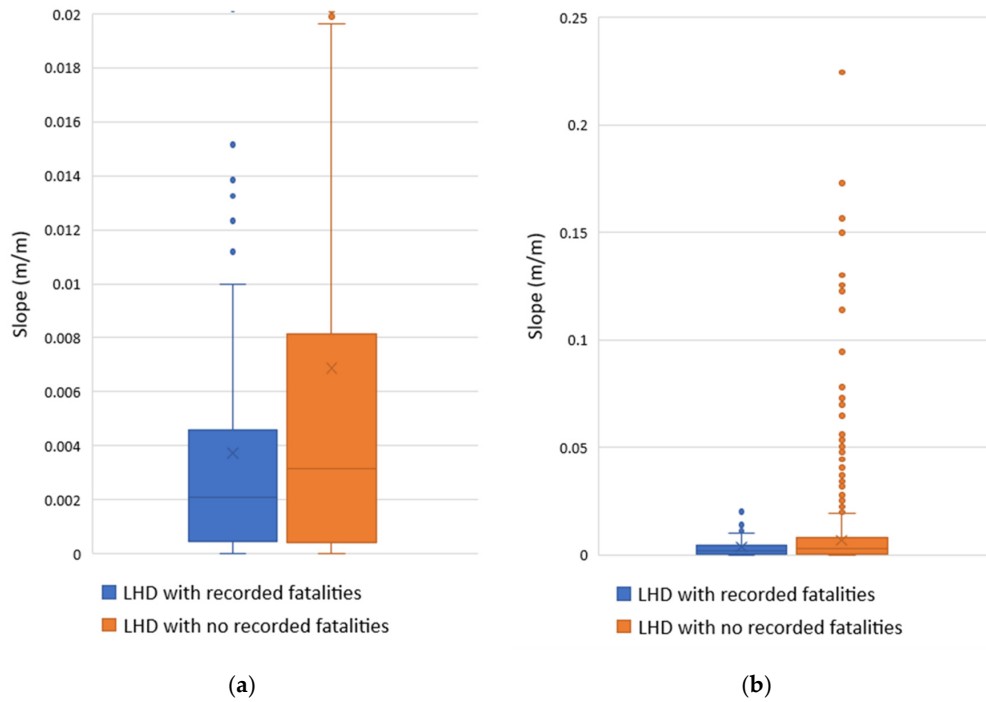

(a)                      (b)

**Figure 9.** Stream slope at low head dams (LHDs) in the sample states: (**a**) Box and whisker plot cutting off the outlier points for low head dams with no recorded fatalities (**b**) The same plot showing the whole range of outlier points for both distributions.

### 3.1.2. Pennsylvania

Each state had slightly different methods of collection, so in order to control for those variables and others, such as the general state topography, individual statistical testing was pursued. In this results section, only the distribution of Pennsylvania is shown for sake of brevity and significance. Pennsylvania has the highest number of low head dams with a recorded fatality of 31, with Colorado and Indiana being the next highest at 12. Colorado and Indiana histograms and box and whisker plots are included in Appendix A (see Figures A1–A6). North Carolina, Maryland, and Idaho had only five dams with recorded fatalities, and New Mexico only had two. The PA, CO, and IN datasets can more reasonably meet all of the assumptions needed for the Mann–Whitney U test [17], because there is enough shape to the fatal LHD distribution to justify the same underlaying distribution assumption to the no recorded fatality distribution. The other four states (NC, MD, ID, and NM) were still included in the statistical testing, but because of this sample size issue, not much time will be spent discussing their results.

Pennsylvania's fatal low head dam stream slope distribution has very similar patterns to the distribution of all of the sample states' fatal low head dam stream slopes: right skewed, and heavily favoring the first bin from 0 to 0.0005 m/m (see Figures 7 and 10). In contrast to it however, Pennsylvania's distribution of low head dams with no recorded fatalities has a bit more of a gradual descent from the first bin on than all of the sample states combined (Figures 8 and 11).

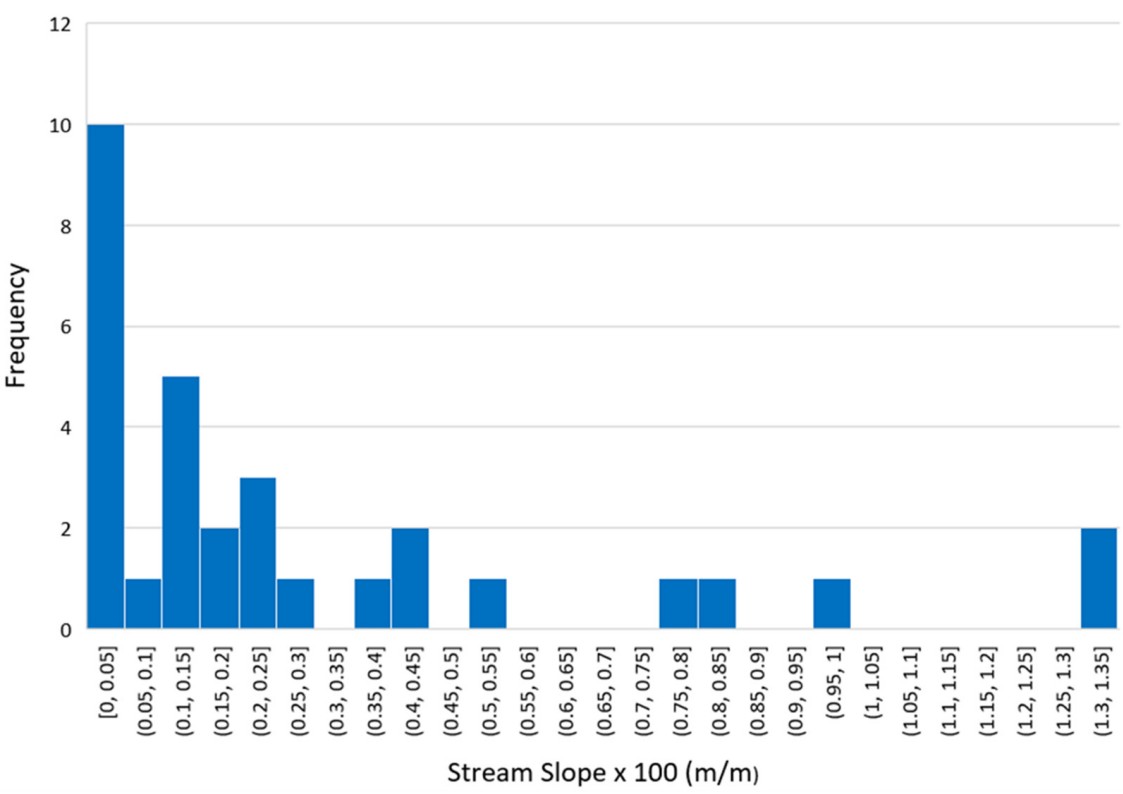

**Figure 10.** Stream slope at low head dams in Pennsylvania with fatalities recorded.

Figure 12 shows the box and whisker plot for each distribution for Pennsylvania's low head dams. The low head dams with fatalities dataset median is about 0.0015 m/m and low head dams without fatalities is about 0.0026 m/m.

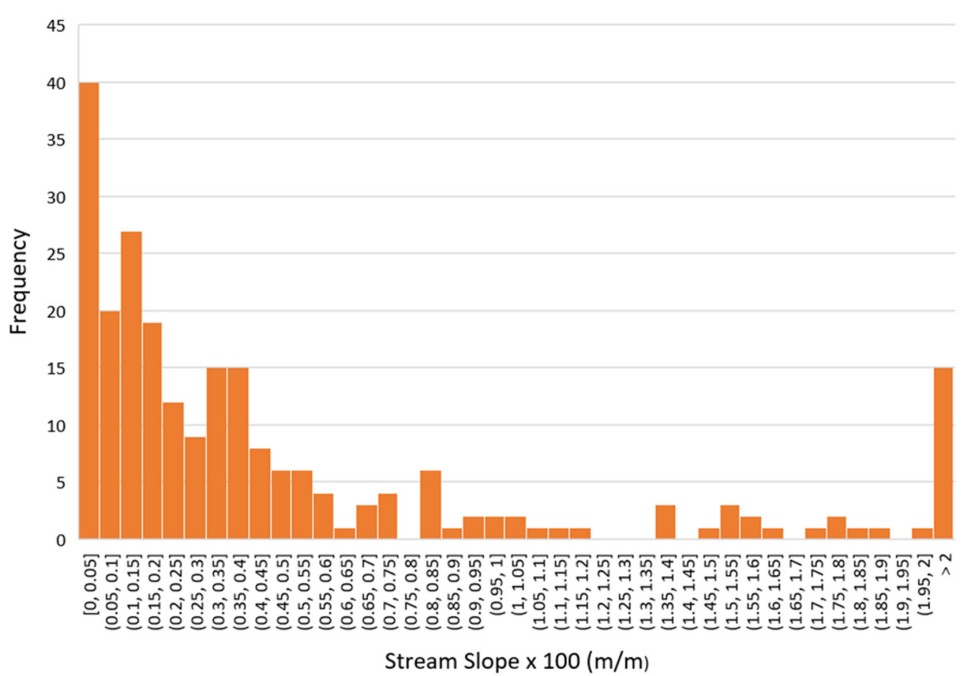

**Figure 11.** Stream slope at low head dams in Pennsylvania with no fatalities recorded.

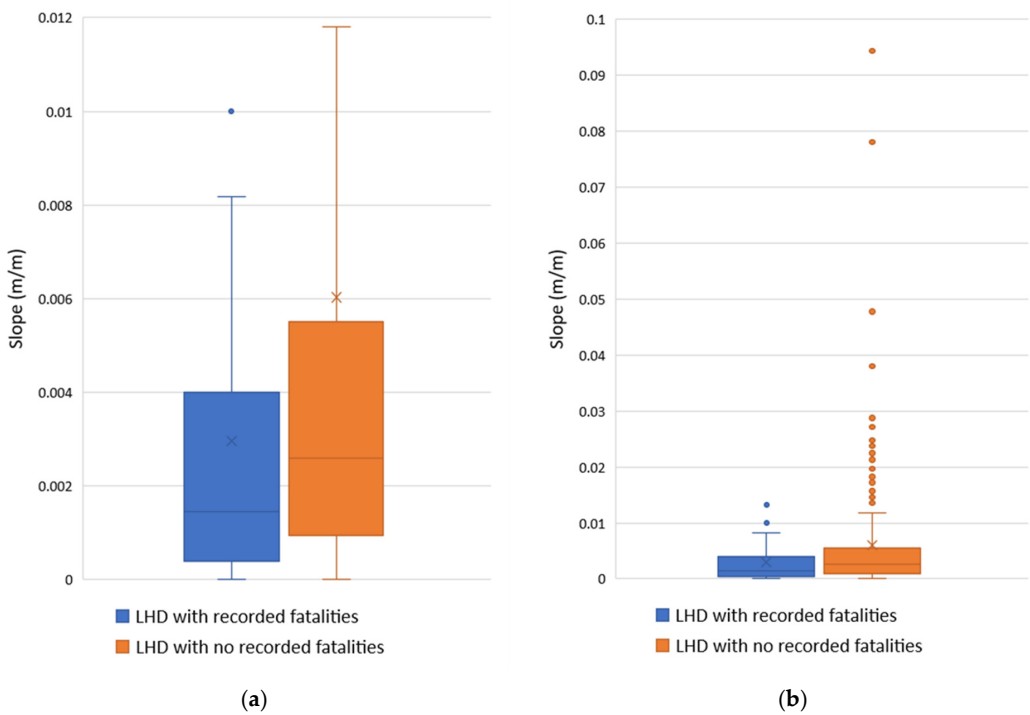

(**a**)    (**b**)

**Figure 12.** Stream slope at low head dams (LHDs) in Pennsylvania: (**a**) Box and whisker plot cutting off the outlier points for low head dams with no recorded fatalities (**b**) The same plot showing the whole range of outlier points for both distributions.

### 3.2. Statisitical Testing

Mann–Whitney U Test Results and Sample Size

The two-tailed Mann–Whitney U test gave a *p*-value of 0.08 for Pennsylvania, 0.44 for Indiana, 0.45 for Colorado, and 0.21 for all sample states combined (see Table 1). None of these or any other fell below the 0.05 significance level set for this study. There was not enough evidence to reject the null hypothesis, which is that there is no significant

difference between stream slope at low head dams with recorded fatalities and those with no recorded fatalities.

**Table 1.** *p*-values from the Mann–Whitney U test and sample size (n) for each dataset.

| State | *p*-Value [1] | n for LHDs with Fatalities | n for LHDs with No Recorded Fatalities |
|---|---|---|---|
| PA | 0.08 | 31 | 236 |
| CO | 0.45 | 12 | 1082 |
| IN | 0.44 | 12 | 148 |
| NC | 0.65 | 5 | 170 |
| MD | 0.37 | 5 | 94 |
| ID | 0.10 | 5 | 562 |
| NM | 0.67 | 2 | 289 |
| All Sample States | 0.21 | 72 | 2581 |

Note: [1] A *p*-value under 0.05 indicates a statistically significant difference.

## 4. Discussion and Recommendations

Results do not disprove the theory that there is some link between flat slopes and a greater potential for danger. Separating out low head dams based on whether they have had a fatality may not be a good enough indication of the presence of a submerged hydraulic jump. This is likely for two reasons. First, the fatality database is incomplete. According to experts, there are far more fatalities than those in the BYU database [1,16]. Second, even with a complete fatality record, submerged hydraulic jumps could occur at low head dams with no recorded fatalities. The goal of the experiment is to be able to identify that exact type of low head dam.

For the data currently available, the best method for building a data-based inventory of hazardous low head dams is to use a method that surveys individual dams, such as the spreadsheet by Wahl [5]. Weir length, structural height of dam above the tailwater invert, stream slope, Manning's n, channel width, and flow ranges for the stream can all be estimated remotely from satellite imagery, the NHDPlusHR [12], the USGS guide to selecting a Manning's roughness coefficient [19], and optionally the flow data from the USGS. The inputs do not need to be fully accurate to identify dangerous dams because low head dams almost always create submerged hydraulic jumps for a large range of their flow conditions [5]. Testing on how much uncertainty in each measurement affects the flow range should be pursued further if this inventory method is undertaken. By comparing flow ranges that are likely to result in a submerged hydraulic jump with annual flow patterns in the stream from the USGS, an idea for how likely it is and when the low head dam is potentially dangerous could be formulated.

It would be beneficial to conduct new tests using field data or other verified information to assess the accuracy of the NHDPlus HR stream slope calculations, such as the ones performed by Cohen et al. on the NHDPlus version 2.0 [14]. The study in this paper could also be performed again using the new Hydrofabric dataset as the input for stream slope at each low head dam [15]. The stream slope calculated in the Hydrofabric should also be tested for accuracy.

**Author Contributions:** Conceptualization, J.W.P. and R.H.H.; methodology, software, and validation, J.W.P.; formal analysis, J.W.P. and R.H.H.; investigation, resources, data curation, writing—original draft preparation, J.W.P.; writing—review and editing, R.H.H.; visualization, J.W.P.; supervision, R.H.H.; project administration, and funding acquisition, R.H.H. All authors have read and agreed to the published version of the manuscript.

**Funding:** This research was funded by the Kenneth and Ruth Wright Family Foundation.

**Informed Consent Statement:** Not applicable.

**Data Availability Statement:** Publicly available datasets were analyzed in this study. This data can be found here: https://apps.nationalmap.gov/downloader/ (accessed on 24 January 2022).

**Acknowledgments:** We acknowledge the work of the Task Force to Create a National Inventory of Low-head Dams and Tony Wahl for his spreadsheet development.

**Conflicts of Interest:** The authors declare no conflict of interest. The funders had no role in the design of the study; in the collection, analyses, or interpretation of data; in the writing of the manuscript; or in the decision to publish the results.

## Appendix A

Colorado Results:

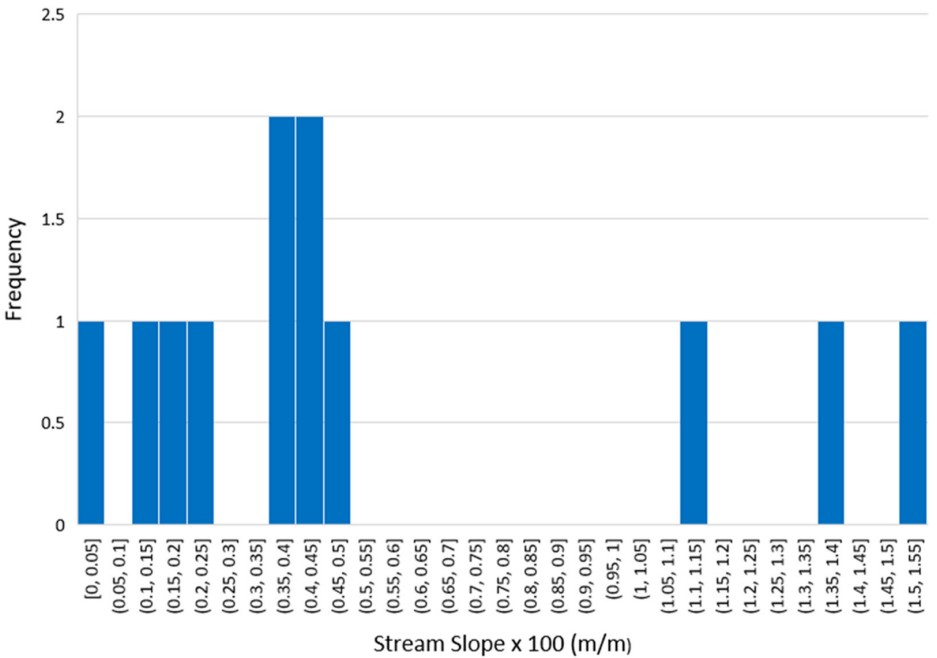

**Figure A1.** Stream slope at low head dams in Colorado with fatalities recorded.

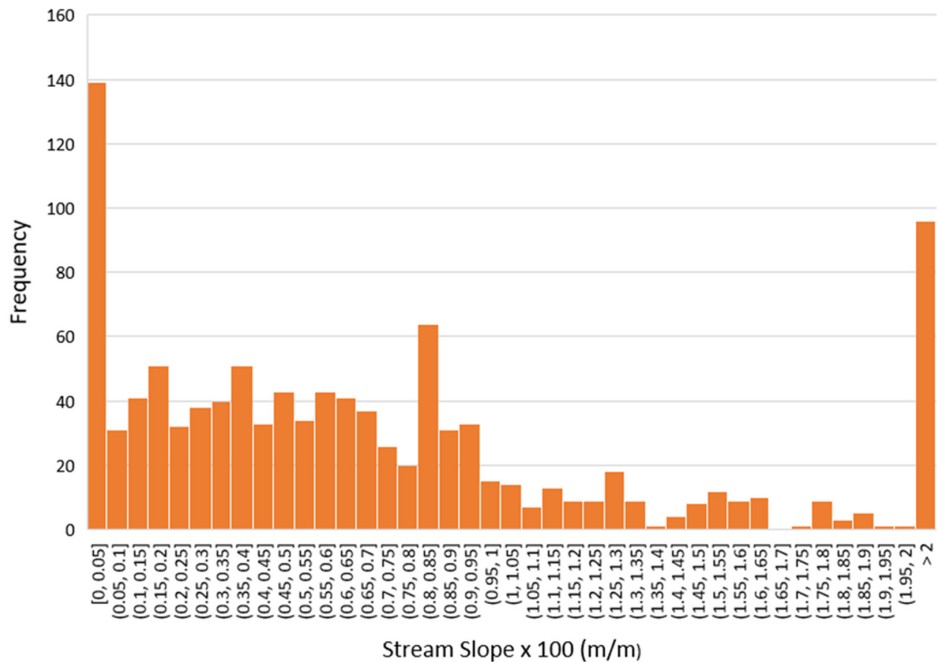

**Figure A2.** Stream slope at low head dams in Colorado with no fatalities recorded.

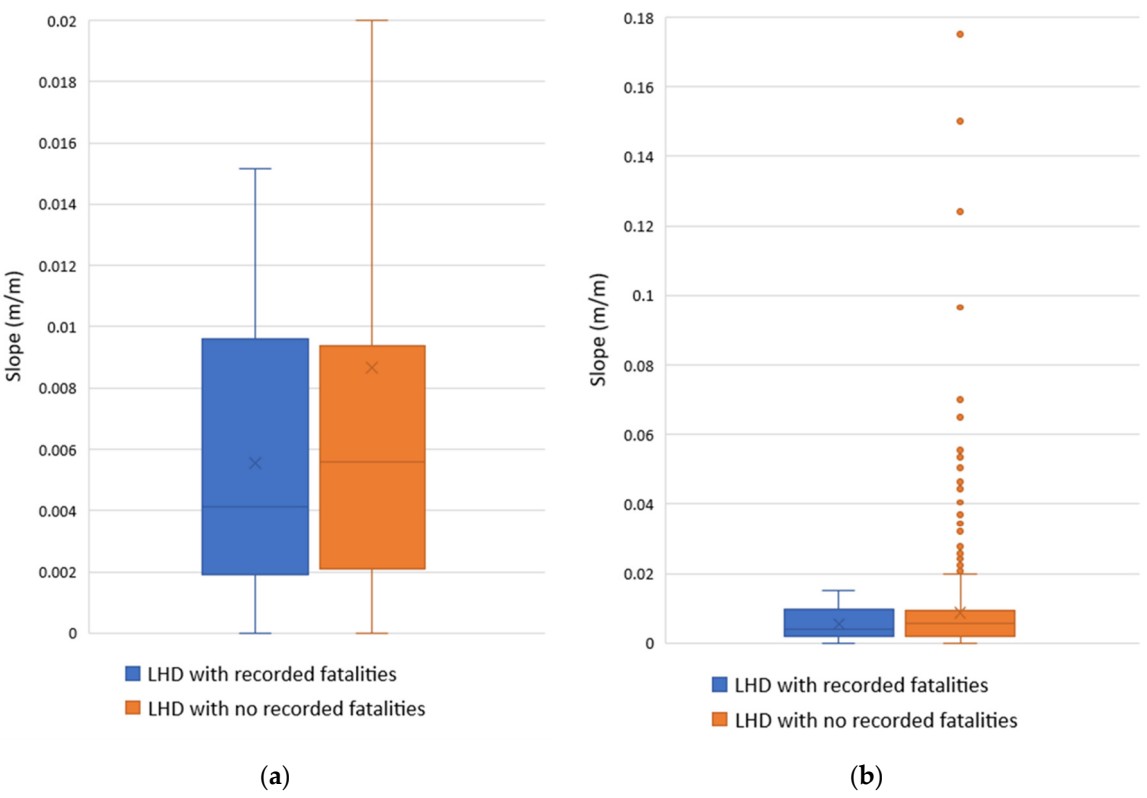

**Figure A3.** Stream slope at low head dams (LHDs) in Colorado: (**a**) box and whisker plot cutting off the outlier points for low head dams with no recorded fatalities (**b**) The same plot showing the whole range of outlier points for both distributions.

Indiana Results:

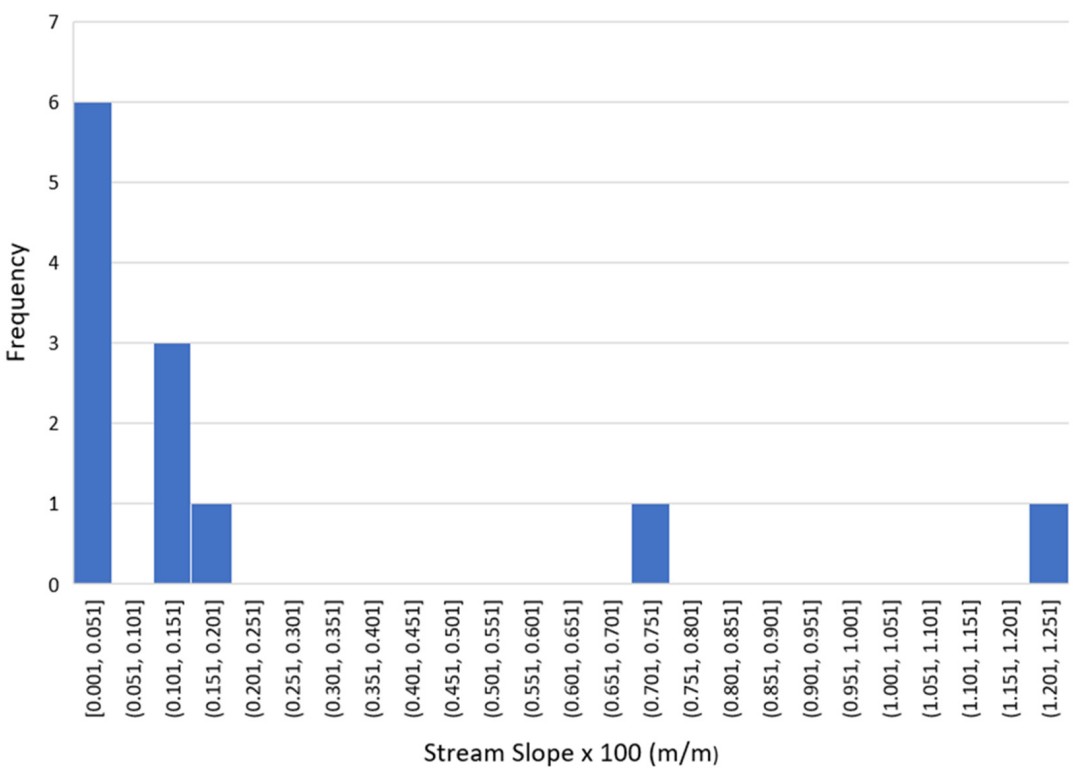

**Figure A4.** Stream slope at low head dams in Indiana with fatalities recorded.

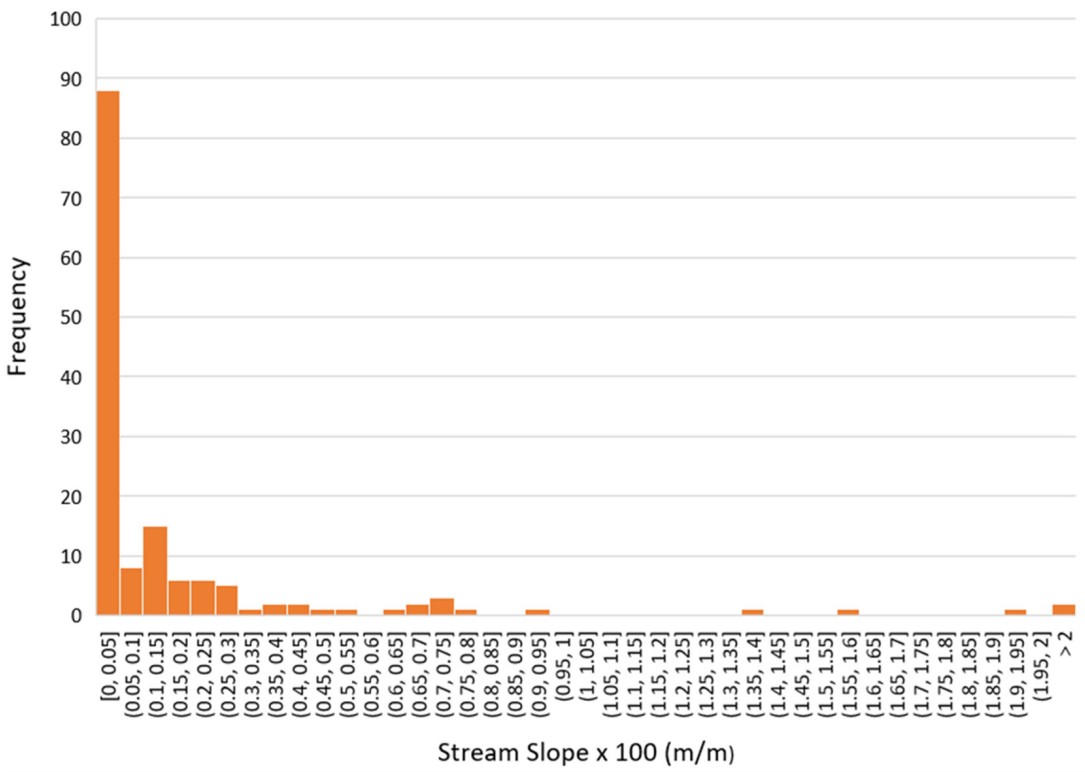

**Figure A5.** Stream slope at low head dams in Indiana with no fatalities recorded.

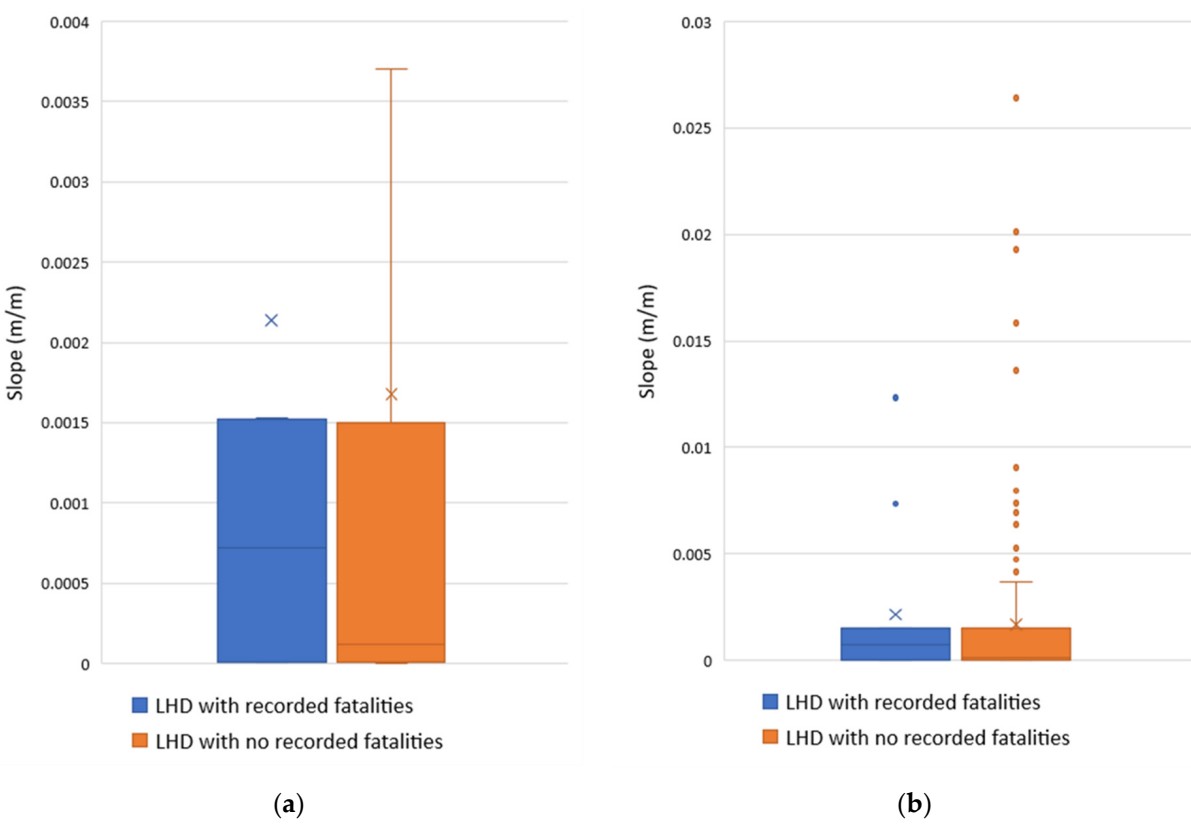

(**a**)                                                                 (**b**)

**Figure A6.** Stream slope at low head dams (LHDs) in Indiana: (**a**) box and whisker plot cutting off the outlier points for low head dams with no recorded fatalities (**b**) The same plot showing the whole range of outlier points for both distributions.

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
