# Peer review of "Stream Slope as an Indicator for Drowning Potential at Low Head Dams"

_water, doi:10.3390/w15030512_

Round 1

Reviewer 1 Report

Review report for paper

Paper entitled “

 Stream slope as an indicator for drowning potential at low head 2 dams ‘’

General remarks:

·      Extensive editing of English language required

·      Quality of all figures should be improved (600dpi is proposed)

Specific remarks

·      The methodology section should be more explained

·      Figure 2 should be improved; the text in the figure is not clear and for its quality 600dpi is proposed

·      It is preferred to add a stream slope map to enrich the discussion of the results

and the significance of your findings

Author Response

  • English: I went through the paper and corrected a couple of minor mistakes. All revisions are tracked in the new version. Otherwise, as native English speakers and experienced writers, we are very confident of our grammar. 
  • All figures have been updated to 600 DPI quality. I increased the resolution for Figure 2, so now all text should be clear.
  • I added a stream slope map figure for the state Pennsylvania as an example of the distribution of stream slope in a state. It is now Figure 5.
  • Methodology section. I went through this section again, and in my opinion I have included all steps necessary for someone to reproduce what I have done. Specific feedback on exactly what steps need to be explained more fully would be very helpful.

Reviewer 2 Report

Although the paper shows usual hydraulic facilities, i.e., their description with explanation (in the first part), the topic is essential. Small dams should be investigated, as well as presented in the literature. Such is a lack because small hydropower plants use such hydro-technical constructions. Without real research on the case study locations, the such task could be done superficially. 

The literature review with the background is at the required level. The methodology is well structured. The paper deserves to be published. Research has a lot of practical aspects with regard to the scientific base. 

I am proposing a minor revision. The authors didn't comment on the impact of the particular type of soil before, under, and after the low dams. What about the deposits and sedimentation? 

Such should be commented on. 

Author Response

  • Thank you for your comments!
  • Impact of soil type before, under, and after dams: We addressed deposits downstream of the dam and how scour and deposition can force high tailwater, and explained that the theory investigated does not account for this circumstance. Because it is the downstream conditions that influence the formation of a submerged hydraulic jump, any upstream sedimentation will not affect the formation of a submerged hydraulic jump. Investigating if soil type under the dams and in the channel has any effect of the formation of submerged hydraulic jumps and the danger of low head dams would be a different investigation, and beyond the scope of this paper. The purpose of the paper is to provide a simple and data-backed method for quickly assessing the risk of low head dams. It is an interesting idea that should be explored, especially in connection with scour holes.